# Stabilizing the Baseline: Reference Gene Evaluation in Three Invasive *Reynoutria* Species

**DOI:** 10.3390/ijms26178265

**Published:** 2025-08-26

**Authors:** Marta Stafiniak, Wojciech Makowski, Adam Matkowski, Monika Bielecka

**Affiliations:** 1Department of Pharmaceutical Biology and Biotechnology, Faculty of Pharmacy, Wroclaw Medical University, Borowska 211A, 50-556 Wroclaw, Poland; fitoterapia@tlen.pl; 2Department of Botany, Physiology and Plant Protection, Faculty of Biotechnology and Horticulture, University of Agriculture in Kraków, 29 Listopada 54, 31-425 Kraków, Poland; wojciech.makowski@urk.edu.pl

**Keywords:** housekeeping genes, invasive species, qRT-PCR normalization, reference-gene validation, *Reynoutria*

## Abstract

Accurate normalization is crucial for reliable gene expression quantification and depends on stably expressed housekeeping genes (HKGs) as internal controls. However, HKGs expression varies with developmental stage, tissue type, and treatments, potentially introducing bias and compromising data accuracy. Thus, validating candidate reference genes under defined conditions is essential. *Reynoutria*, also known as giant Asian knotweeds, is a Polygonaceae family genus of several medicinal plants producing a diverse array of specialized metabolites of pharmacological interest. Outside their native range, these plants are also noxious invasive weeds, causing significant environmental and economic threats. Research on stable reference genes in these species is limited, with a primary focus on *R. japonica*. To enable accurate gene expression analysis related to specialized metabolism and natural product biosynthesis, we aimed to identify the most stable reference genes across the most common species: *R. japonica* Houtt., *R. sachalinensis* (F. Schmidt) Nakai, and their hybrid—*R.* × *bohemica* Chrtek & Chrtková. In this study, we evaluated twelve candidate HKGs (*ACT*, *TUA*, *TUB*, *GAPDH*, *EF-1γ*, *UBQ*, *UBC*, *60SrRNA*, *eIF6A*, *SKD1*, *YLS8*, and *NDUFA13*) across three tissue types (rhizomes, leaves, and flowers) from three *Reynoutria* species sampled at peak flowering. Primer specificity and amplification efficiency were confirmed through standard-curve analysis. We assessed expression stability using ΔCt, geNorm, NormFinder, and BestKeeper, and generated comprehensive rankings with RefFinder. Our integrated analysis revealed organ- and species-dependent stability differences, yet identified up to three reference genes suitable for interspecific normalization in *Reynoutria*. This represents the first systematic, comparative validation of HKGs across closely related knotweed species, providing a robust foundation for future transcriptomic and functional studies of their specialized metabolism and other biological processes.

## 1. Introduction

Gene expression profiles serve as key indicators of cellular state, with shifts in transcript abundance reflecting adaptive responses to developmental cues or environmental stimuli. Quantitative real-time PCR (qRT-PCR, qPCR) is broadly utilized to investigate gene function due to its high sensitivity, accuracy, and adaptability in molecular biology applications [1,2]. It has become a go-to approach for detecting and quantitatively analyzing target gene expression, thereby playing a crucial role in functional gene expression research in plants. Although genome and transcriptome sequencing also provide insights into gene expression, qRT-PCR remains particularly favored due to its superior sensitivity, efficiency, and adaptability to various experimental conditions [3,4,5]. However, the accuracy of qRT-PCR results can be significantly influenced by factors such as sample quality, primer specificity, and amplification efficiency [4,6,7]. Proper experimental design, including the selection of appropriate endogenous controls and data analysis methods, is crucial for obtaining reliable results [8,9].

Due to their relatively stable expression, housekeeping genes (HKGs) are routinely used as internal reference genes [10,11,12,13,14,15], yet their expression can vary markedly across different tissues and species [16]. HKGs are genes expressed constitutively across cells and organisms, performing essential cellular functions, such as energy generation, substance synthesis, cell death, and cell defense. Reliable normalization using HKGs enables the detection of biologically relevant changes in target genes, which is essential in studies spanning developmental biology and stress responses [17].

The genus *Reynoutria* Houtt. (*Polygonaceae*) comprises six species and two natural hybrids, of which *R. japonica* Houtt., *R. sachalinensis* (F. Schmidt) Nakai, and their hybrid *R.* × *bohemica* Chrtek & Chrtková are among the most aggressive invaders worldwide. Japanese knotweed (*R. japonica*) is native to Japan, Korea, and China and colonizes riverbanks, roadsides, and volcanic slopes [18]. *R. sachalinensis* (Giant knotweed) occurs on Sakhalin, Hokkaido, and Honshu at lower elevations [19]. Bohemian knotweed (*R.* × *bohemica*) arises from pollination of male-sterile *R. japonica* by *R. sachalinensis* pollen, resembles *R. japonica* morphologically, and is often misidentified. All three reproduce vigorously via rhizomes, exhibit rapid vegetative spread, and are classified as invasive aliens in Europe, North America, and New Zealand [18,20,21]. Only *R. japonica* is listed in the European Pharmacopoeia as “*Polygoni cuspidati rhizoma et radix*” due to its emodin and piceid content, and it has a long history in Chinese and Japanese traditional herbal medicine [22]. *R. sachalinensis* has limited folk use for treating joint pain and jaundice [23], whereas *R.* × *bohemica* has been primarily studied for its biomass applications [24]. Phytochemical profiling of the three taxa revealed over 130 detected metabolites [25], with *R.* × *bohemica* more similar to *R. japonica* than to *R. sachalinensis*, suggesting its potential as an alternative source of bioactive compounds. However, distinguishing between *Reynoutria* species and hybrids remains a critical issue in both research and utilization of these plants; hence, our previous publication was dedicated entirely to this issue and provided a comprehensive key for distinguishing *Reynoutria japonica*, *R. sachalinensis*, and *R.* × *bohemica* based on morphological features, metabolic profiles, and molecular markers [26]. Additionally, in light of the existing variety of approaches used in studying invasive species and the genetic phenomena associated with invasiveness, we examined the population structure and species identification and proposed molecular markers to track these processes. Our results highlighted that morphologically similar plants may differ considerably at the genetic and phytochemical levels, underscoring the importance of molecular characterization to avoid misinterpretation.

Within the genus *Reynoutria*, recent research has primarily employed next-generation sequencing to analyze chloroplast and mitochondrial genomes as well as molecular markers, thereby elucidating phylogenetic relationships and genetic diversity [26,27,28,29,30]. Only a limited number of studies have addressed suitable reference genes for gene expression normalization in these species. To date, for *R. japonica*, *NADH dehydrogenase [ubiquinone] 1 alpha subcomplex subunit 13-A* (*NDUFA13*) and *elongation factor 1-gamma* (*EF-1γ*) have been identified as stable reference genes across different tissues and stress conditions, both propagated and performed in vitro [31]. In more phylogenetically distant *Pleuropterus multiflorus* (Thunb.) Turcz. ex Nakai (syn. *Polygonum multiflorum*), *protein phosphatase 2A* (*PP2A*) [32,33], *GAPDH* [33], as well as *polyubiquitin 14* (*UBQ14*), *polyubiquitin 4* (*UBQ4*-1), *and S-adenosylmethionine synthase* (*SAMS*) [34] exhibited the highest stability. This significant knowledge gap underscores the pressing need for identifying universally stable or interspecific HKGs. Developing a unified set of validated reference genes would significantly enhance the accuracy and comparability of gene expression studies, facilitating insights into genetic regulation and aiding comparative molecular studies within this biologically and pharmacologically complex genus.

This study aimed to identify highly stable reference genes for qRT-PCR normalization across three invasive *Reynoutria* species. Twelve candidate genes—*actin* (*ACT*), *α-tubulin* (*TUA*), *β-tubulin* (*TUB*), *glyceraldehyde-3-phosphate dehydrogenase* (*GAPDH*), *elongation factor 1-γ* (*EF-1γ*), *ubiquitin domain–containing protein* (*UBQ*), *ubiquitin-conjugating enzyme* (*UBC*), *60S ribosomal RNA* (*60SrRNA*), *eukaryotic translation initiation factor 6A* (*eIF6A*), *suppressor of K^+^ transport defect1* (*SKD1*), *thioredoxin-like protein* (*YLS8*), and *NADH dehydrogenase [ubiquinone] 1 α-subcomplex subunit 13-A* (*NDUFA13*) were selected from transcriptomic data as described by Wang et al. [31] and assayed by qRT-PCR in rhizome, leaf and flower tissues of four representatives of three species: one *R. japonica*, two *R.* × *bohemica* and one *R. sachalinensis*. Expression stability was evaluated using the comparative ΔCt method [35], geNorm [36], NormFinder [37], BestKeeper [38], and the integrative RefFinder algorithm [39,40], and the accuracy of results was validated by analyzing the expression patterns of the *PAL* and *CHS* genes.

To our knowledge, this study is the first systematic attempt to identify a universally stable housekeeping genes among the commonly compared *Reynoutria* species, laying the groundwork for future transcriptomic and functional studies in this chemically rich yet underexplored genus.

## 2. Results

### 2.1. Primer Specificity and Amplification Efficiency

Melt-curve analysis revealed a single peak for each candidate reference gene, indicating highly specific amplification (Figure 1A). Agarose gel electrophoresis confirmed that each primer pair produced a single amplicon of the expected size (75–240 bp) (Figure 1B,C). No non-specific bands or primer dimers were detected.

Standard-curve assays generated slopes, amplification efficiencies (E), and correlation coefficients (R^2^) within acceptable limits: E ranged from 100% (*UBQ* and *eIF6A*) to 105% (*CHS*), meeting the 90–110% criterion, while R^2^ values spanned 0.99–0.999 (all > 0.98). These results demonstrated that all primer sets have suitable specificity and efficiency for subsequent qRT-PCR analyses. Primer sequences, amplicon lengths, empirically established efficiencies, and R^2^ values are provided in Table 1.

### 2.2. Expression Profiles of Candidate Reference Genes

The consistency of expression among the twelve candidate reference genes across various tissues and species was assessed through their cycle threshold (Ct) values obtained via qRT-PCR. Given that Ct values are inversely proportional to transcript abundance, genes exhibiting lower Ct values are indicative of higher expression levels.

Ct values across all samples ranged from 13.49 to 35.12. The mean Ct values varied from 19.53 for *EF-1γ*, indicating high gene expression, to 24.74 for *TUB*, reflecting relatively lower gene expression (Appendix A).

Box plots of Ct distributions (Figure 2) illustrate expression variation across samples. Genes with lower dispersion in Ct values indicate greater expression stability. *EF-1γ* showed the least variation, reflecting consistent expression, while *GAPDH* exhibited the highest variability among the tested reference genes (Figure 2A).

Among the tissue types analyzed, flowers exhibited the lowest Ct values, thereby suggesting the highest overall transcriptional activity. Specifically, 21RS_FL demonstrated the lowest average Ct value of 16.45. In contrast, rhizomes, particularly those from *R. sachalinensis*, displayed the highest mean Ct value of 33.78, consistent with the reduced transcriptional activity typically associated with lignified underground tissues, while the leaves showed intermediate values. Mean expression in flowers was the most consistent and stable across species (Figure 2B).

### 2.3. Expression Stability Assessment

Due to the significant variability in expression levels of candidate reference genes across different samples, statistical analysis was employed to evaluate and rank their expression stability, as well as to determine the optimal number of reference genes needed for reliable gene expression analysis.

#### 2.3.1. ΔCt Method

The stability of each reference gene was assessed using the ΔCt method, where genes with the lowest average standard deviation across samples are considered the most stable (Figure 3). The analysis showed that *60SrRNA* consistently exhibited the least variation in expression, making it the most stable gene overall across all *Reynoutria* samples. In specific organs, *60SrRNA* was the most stable in leaves and rhizomes. However, in rhizomes, all candidate genes showed high ΔCt values, indicating significant variability and a lack of reliable reference genes in this tissue type. In flowers, *SKD1* was the most stable. When analyzed by species, *60SrRNA* remained the most stable gene in *R. japonica*, while *UBC* was highest in *R.* × *bohemica*, and *eIF6A* in *R. sachalinensis*. Yet, similar to rhizomes, *R. sachalinensis* samples displayed increased variability across all candidate genes, which undermines confidence in expression stability for this species.

#### 2.3.2. geNorm Analysis

The stability of the twelve candidate reference genes was assessed using the geNorm algorithm, which calculates an average expression stability value (M) based on pairwise comparisons among all tested genes. According to geNorm, lower M values indicate higher expression stability, with values below 1.5 considered acceptable for use as reference genes [36]. In this study, *ACT*, *SKD1*, *60SrRNA*, *GAPDH*, *NDUFA13*, and *EF-1γ* consistently exhibited M values below 1.5 across all samples, confirming their suitability as reference genes in gene expression analyses across different species and tissue types (Figure 4). For leaves, flowers, and *R.* × *bohemica*, all twelve genes demonstrated M values below the 1.5 threshold. In contrast, in variable sample types such as rhizomes and *R. sachalinensis* organs, only a subset of genes met this criterion, with *ACT* and *SKD1* emerging as the most stable and reliable reference genes.

Besides ranking gene stability, geNorm was employed to determine the optimal number of reference genes needed for precise normalization. This involved calculating the pairwise variation (V_n_/V_n+1_) between consecutive normalization factors (NF), with a common cutoff of V < 0.15 indicating whether adding more genes significantly improves normalization (Appendix A) (Figure 5). For example, in *R.* × *bohemica*, where V3/4 = 0.1527 and V4/5 = 0.1316, the result suggests that four reference genes are sufficient, as adding a fifth yields only a marginal improvement. Conversely, in leaves, flowers, and *R. japonica* samples, V2/3 < 0.15, supporting the use of just two reference genes for optimal normalization.

#### 2.3.3. NormFinder Analysis

To validate the results obtained from geNorm, the stability of candidate reference genes was re-evaluated using NormFinder. This model-based algorithm calculates a gene expression stability value (S) for each gene by evaluating both intra- and inter-group variation [37]. Prior to analysis, Ct values are transformed into relative expression levels, as required by the algorithm. Lower S values indicate higher stability and better suitability as reference genes. Unlike geNorm, which relies on pairwise comparisons, NormFinder employs an analysis of variance (ANOVA) approach to estimate variability and directly pinpoint the most stable genes across sample groups.

The stability analysis using NormFinder revealed notable variation in reference gene performance across different *Reynoutria* tissues and species (Figure 4). When evaluating all samples collectively, *eIF6A* (S = 3.65) and *60SrRNA* (S = 3.69) emerged as the most stable genes, indicating their potential as broadly applicable reference genes (Appendix A). In leaf tissue, *GAPDH* (0.336) and *YLS8* (0.338) showed the highest stability, making them well-suited for normalization in foliar expression studies. In rhizomes, although overall stability values were higher—reflecting greater variability—*eIF6A* (5.33), *60SrRNA* (6.15), and *SKD1* (7.06) ranked as the most stable. For flowers, *SKD1* (0.069) exhibited exceptional stability, followed closely by *TUB* (0.093) and *GAPDH* (0.207). In *R. japonica* samples, both *SKD1* and *EF-1γ* (0.017) demonstrated outstanding stability, while *UBC* (0.280), *60SrRNA* (0.365), and *eIF6A* (0.453) were most stable in *R. × bohemica*. Conversely, *R. sachalinensis* showed the least stability overall, with *eIF6A* (6.23), *NDUFA13* (8.74), and *SKD1* (9.01) ranking highest within a relatively unstable set. Notably, *TUB* and *YLS8* consistently showed poor stability across multiple tissues and species, indicating they are unsuitable as reference genes.

#### 2.3.4. BestKeeper Analysis

BestKeeper evaluates reference gene stability by calculating the standard deviation (SD) and coefficient of variation (CV) of Ct values, where lower values indicate greater stability. Additionally, it computes Pearson correlation coefficients (r) between each gene and the BestKeeper index, defined as the geometric mean of all candidate reference genes [38]. This descriptive statistical approach offers a complementary perspective to geNorm and NormFinder, providing further validation for reference gene selection.

In this study, BestKeeper was used to assess the stability of twelve candidate reference genes across multiple *Reynoutria* tissues and species (Figure 6). The analysis revealed that *SKD1* (SD = 3.57; CV = 16.05%) and *EF-1γ* (SD = 3.58; CV = 17.24%) were the most stable genes overall (Appendix A). The *60SrRNA* gene also showed low variability, supporting its suitability for normalization. In contrast, *TUB* (SD = 5.85; CV = 25.78%) and *TUA* (SD = 5.53; CV = 28.40%) were identified as the least stable genes across all samples due to their high coefficient of variation (CV).

Gene stability varied notably across organs. In leaf samples, *EF-1γ* (SD = 0.65; CV = 3.26%) and *SKD1* (SD = 0.68; CV = 3.11%) exhibited the highest stability, while *TUB* (SD = 2.48; CV = 9.38%) was the least stable. In rhizomes, *60SrRNA* (SD = 3.83; CV = 14.64%) was the most stable, whereas *TUB* (SD = 10.92; CV = 50%) and *YLS8* (SD = 10.45; CV = 100%) displayed extreme variability and were unsuitable for normalization. Flower tissues analysis showed *EF-1γ* (SD = 0.49; CV = 2.94%) and *UBQ* (SD = 0.50; CV = 2.58%) as the most stable genes. Although *NDUFA13* (SD = 0.85; CV = 4.82%) and *UBC* (SD = 0.86; CV = 4.68%) were the least stable in flowers, overall variability in this tissue was relatively low.

Species-specific analysis further underscored the need for tailored reference gene selection. In *R. japonica*, *UBQ* (SD = 2.35; CV = 10.50%) and *GAPDH* (SD = 2.88; CV = 15.35%) were the most stable genes, while *YLS8* (SD = 7.95; CV = 66.67%) was highly unstable. For *R. × bohemica*, *EF-1γ* (SD = 1.71; CV = 8.87%) and SKD1 (SD = 2.15; CV = 10.26%) exhibited strong stability, whereas *NDUFA13* and *TUA* (SD = 3.71 and 3.87, respectively; CV ≈ 17–18%) were the least stable. In *R. sachalinensis*, *eIF6A* (SD = 4.91; CV = 19.22%) and *60SrRNA* (SD = 5.59; CV = 23.36%) showed the highest stability, while *TUB* (SD = 10.83; CV = 66.67%) was again the most variable and unsuitable as a reference gene.

Overall, the BestKeeper analysis confirms that the stability of reference genes is highly dependent on organ type and species. *SKD1*, *EF-1γ*, and *60SrRNA* consistently ranked among the most stable genes across multiple conditions and are recommended by BestKeeper for normalization in *Reynoutria* gene expression studies. Conversely, *TUB* and *TUA* exhibited high variability and should be avoided.

#### 2.3.5. Comprehensive Stability Ranking

Across all samples, *SKD1* emerged as the most stable gene (Geomean = 1.57), making it the top choice for general normalization across *Reynoutria* tissues and species (Table 2) (Appendix A). It was closely followed by *60SrRNA* (2.06) and *ACT* (2.78), both of which are also strong candidates for broad-scale applications. In contrast, *YLS8* (11.47) and *TUB* (11.24) were the least stable overall and should be avoided in most contexts due to inconsistent expression.

Stability patterns varied by organ type. In leaves, *60SrRNA* (2.63) was the most stable gene, followed by *EF-1γ* (3.15) and *YLS8* (3.16). In rhizomes, *60SrRNA* (1.68) again ranked highest, followed by *SKD1* (2.45) and *NDUFA13* (2.78). In flowers, *SKD1* (2.43) demonstrated the highest stability, followed by *TUB* (2.45) and *60SrRNA* (2.71), which also ranked well, underscoring the variation in *TUB* stability across tissues.

Species-specific analysis further confirmed these trends. In *R. japonica*, *SKD1* (2.43) was most stable, followed by *EF-1γ* (2.63) and *60SrRNA* (2.78). In *R. × bohemica*, *UBC* (2.06) ranked highest, while *SKD1* (2.45) and *eIF6A* (2.55) also performed well. For *R. sachalinensis*, *eIF6A* (1.63) was the most stable, with *SKD1* (2.45) and *60SrRNA* (2.51) close behind.

### 2.4. Evaluation of Candidate Reference Genes for Expression Analysis in Reynoutria

To assess the reliability of the selected reference genes, we analyzed the expression of *PAL* (*phenylalanine ammonia-lyase*) and *CHS* (*chalcone synthase*) across different tissues of three *Reynoutria* species. These genes are key to the phenylpropanoid and flavonoid biosynthesis pathways, respectively, and play crucial roles in plant defense and the production of secondary metabolites such as stilbenes and vanicosides—compounds of particular significance in *Reynoutria* species. Based on RefFinder rankings, the three most stable reference genes (*SKD1*, *60SrRNA*, and *ACT*) and the two least stable (*TUB* and *YLS8*) were used to normalize *PAL* and *CHS* expression data. The performance of these reference genes was evaluated to determine their suitability for accurate gene expression analysis in *Reynoutria*.

As shown in Figure 7, the expression patterns of *PAL* and *CHS* were consistent and biologically reasonable when normalized using the stable reference genes *SKD1*, *60SrRNA*, and *ACT*. In contrast, normalization with the unstable reference genes (*TUB* and *YLS8*), either individually or in combination, resulted in markedly altered expression profiles characterized by inflated variability and inconsistent trends. These discrepancies underscored the importance of reference gene stability in interpreting qPCR data. The results highlighted that the use of unreliable reference genes can distort target gene expression and compromise experimental accuracy, further validating the selection of *SKD1*, *60SrRNA*, and *ACT* as appropriate normalizers in *Reynoutria*.

## 3. Discussion

Housekeeping genes (HKGs) are traditionally defined by four primary biological attributes: stable expression across various samples, essentiality for basic cellular functions, participation in fundamental maintenance processes, and evolutionary conservation. They are often characterized as genes necessary for cellular existence regardless of tissue type, developmental stage, or environmental condition [15]. Despite their presumed stability, numerous studies have demonstrated that HKGs expression can vary significantly depending on tissue types, developmental stages, experimental treatments, environmental conditions, and species [41,42,43]. Genes frequently used as internal controls, such as GAPDH or β-actin, have shown inconsistent performance across different experiments, highlighting the critical need to validate their stability for each specific experimental condition empirically [44]. Therefore, rigorous validation of candidate reference genes is now recognized as essential for obtaining reliable and reproducible quantitative PCR results [42,45]. Moreover, studies increasingly recommend employing multiple validated HKGs to ensure robust normalization of qPCR data, thereby enhancing the accuracy of gene expression analysis [46,47].

In this study, we evaluated the expression stability of twelve candidate reference genes across various organs and species within the *Reynoutria* genus using four well-established algorithms: ΔCt, geNorm, NormFinder, and BestKeeper. The results from these methods were combined using the RefFinder platform, which integrates individual rankings into a comprehensive consensus based on a weighted geometric mean. This multi-algorithm approach reduces biases inherent to any single method and provides a robust evaluation of reference gene suitability.

Our analysis revealed significant variation in reference gene stability across species and plant organ types. While no gene remained perfectly consistent under all conditions, SKD1, 60SrRNA, and ACT were identified as the most stable genes when analyzing all samples collectively. These genes were therefore considered most suitable for comparisons between species within *Reynoutria*. Species-specific analyses refined these recommendations further: *SKD1* and EF-*1γ* were the most stable in *R. japonica*, *UBC* and *eIF6A* in *R. × bohemica*, and *eIF6A* and *60SrRNA* in *R. sachalinensis*. Likewise, organ-specific expression profiles identified *60SrRNA* and *EF-1γ* as the best candidates in leaves, *SKD1* and *60SrRNA* in rhizomes, and *SKD1* and *TUB* in flowers.

Each analytical tool provided unique insights into reference gene performance. The ΔCt method, which assesses consistency in Ct differences between gene pairs across samples, showed that *60SrRNA* exhibited the most stable expression across all datasets. This highlights its strong potential as a universal reference gene for *Reynoutria* studies.

The geNorm algorithm, which evaluates pairwise variation and calculates M values to determine the minimum number of genes needed for reliable normalization, supported the use of *SKD1*, *60SrRNA*, and *ACT* despite the V values not meeting the strict 0.15 threshold. *SKD1*, in particular, approached the threshold for acceptable stability, reinforcing its usefulness, especially when combined with other stable genes.

NormFinder, which considers both intra- and intergroup expression variability, further validated *SKD1* and *60SrRNA* as dependable reference genes. *SKD1* showed exceptional stability in both flower tissue and *R. japonica*, while *eIF6A* also ranked high across various conditions. These results highlight NormFinder’s ability to model group-specific effects, offering a complementary perspective to geNorm’s pairwise approach.

A gene identified as highly stable by both geNorm and NormFinder is considered a strong candidate for normalization, as it demonstrates consistent expression in pairwise comparisons and across experimental groups. Conversely, discrepancies between geNorm and NormFinder rankings, such as a gene appearing stable in geNorm but unstable in NormFinder, may reflect bias from group-specific variation that geNorm does not explicitly account for. Therefore, combining the results of both algorithms offers a more detailed and reliable assessment of reference gene stability [48].

BestKeeper analysis, which utilizes standard deviation and coefficient of variation metrics, confirmed the organ- and species-dependent stability of reference genes. While *SKD1* consistently ranks among the top candidates across tissues and species—particularly in leaves, flowers, and *R. × bohemica*—greater variability is observed in rhizomes and *R. sachalinensis*. *EF-1γ* and *60SrRNA* also perform well across multiple contexts. Conversely, *TUB* and *TUA* exhibit high variability and are not recommended for normalization, especially in rhizome samples and *R. sachalinensis*.

The integrated RefFinder results provide practical recommendations tailored to specific experimental contexts. Its accessible interface makes it suitable for researchers with different levels of bioinformatics knowledge and has contributed to its widespread adoption in the qPCR community. The strength of RefFinder lies in its ability to provide a comprehensive consensus ranking using a weighted geometric mean, thereby enhancing confidence in selecting stable reference genes [39,40]. However, it is essential to acknowledge that RefFinder does not introduce novel algorithms, as it solely relies on the performance and limitations of the integrated methods, and the ranking process is akin to a “black box” providing limited transparency and control over how individual rankings are combined, which does not seem to be discussed enough [49].

RefFinder uses a geometric mean to combine rankings from four algorithms, but the exact weighting scheme is not disclosed, making its internal processes opaque. This lack of transparency limits customization, such as adjusting thresholds or accounting for outliers, and makes troubleshooting difficult. While RefFinder offers a useful consensus, its fixed parameters may not suit all datasets. To mitigate the “black box” issue, results should be interpreted within the context of the experiment, and reference gene choice should be validated through independent methods like amplification curve analysis. Consequently, it is advisable to examine the degree of agreement among the constituent algorithms. When a candidate gene is consistently ranked highly across all four methods, it offers strong evidence of proper expression stability. Despite these caveats, RefFinder remains a valuable and widely trusted tool for enhancing the reliability of qPCR normalization through multifactorial evaluation of reference genes [4,16,50,51,52,53].

For general use across all samples, RefFinder indicated *SKD1* as the top candidate, with *60SrRNA* and *ACT* serving as reliable alternatives. For organ-specific studies, *60SrRNA* and *EF-1γ* are suitable for leaves, *SKD1* and *60SrRNA* for rhizomes, and *SKD1* and *TUB* for flowers. In species-specific applications, *SKD1* and *EF-1γ* are ideal for *R. japonica*, *UBC* and *eIF6A* for *R. × bohemica*, and *eIF6A* and *60SrRNA* for *R. sachalinensis*. Genes such as *YLS8*, *UBQ*, and *TUB*, which consistently received low stability rankings, should be avoided to prevent normalization errors.

To validate the selected reference genes, we examined the expression patterns of *PAL* and *CHS*, key enzymes involved in phenylpropanoid and flavonoid biosynthesis. When normalized using the top-ranked genes (*SKD1*, *60SrRNA*, and *ACT*), *PAL* and *CHS* expression levels showed consistent and biologically meaningful trends across different species and tissues. In contrast, normalization with unstable reference genes (*TUB* and *YLS8*) resulted in erratic expression patterns and inflated variability, illustrating how selecting inappropriate reference genes can distort experimental outcomes [36].

Wang et al. [31] conducted the first systematic study of reference gene stability in *Polygonum cuspidatum* (syn. *Reynoutria japonica*), focusing on in vitro-cultivated material and analyzing gene expression in roots, stems, and leaves, since rhizomes do not develop under in vitro conditions. They identified *NDUFA13* and EF-1γ as the most stable reference genes across various tissues and abiotic or hormonal treatments, with additional genes like *GAPDH* and *SKD1* performing well under specific conditions. However, their study was limited to a single species and controlled growth conditions, which may not fully represent the gene expression variability in naturally grown plants. In contrast, our study examined multiple *Reynoutria* species grown under natural field conditions, including various tissues such as rhizomes, flowers, and leaves. This broader sampling uncovered greater differences in reference gene expression among organs, likely due to natural environmental variation, developmental stage differences, and the inherently lower transcriptional activity of rhizomes and woody tissues. Our results consistently identify *60SrRNA* and *SKD1* as stable reference genes across all tested samples, with *EF-1γ* also performing well in several contexts. While we confirm and expand upon some of Wang’s findings—using selected primers and GenBank-deposited sequences from their study—our broader taxonomic and organ-level approach offers a more comprehensive and ecologically relevant assessment of reference gene stability within the *Reynoutria* genus. To the best of our knowledge, this investigation represents the first comprehensive and systematic effort to identify universally stable housekeeping genes across the commonly studied *Reynoutria* species.

In this context, it is worth mentioning that studies identifying interspecies HKGs are rare due to the complexity of gene expression across organisms, tissues, and environments. The lack of universally conserved HKGs, species-specific biology, and limited comparable datasets pose significant challenges. Joshi et al. [15] statistically examined housekeeping genes across species, showing that while housekeeping functions are conserved, specific genes often are not. They highlight the influence of environment, development, and tissue type, and propose redefining “housekeepingness” as a continuous, context-dependent trait rather than a fixed set of genes.

In conclusion, our study provides a comprehensive evaluation of candidate reference genes for precise qPCR normalization in *Reynoutria*. Reference gene stability varied significantly among *Reynoutria* tissues and species, underscoring the importance of context-specific selection. By integrating multiple analytical methods, we identified stable reference genes suitable for general use as well as for species- and organ-specific applications. These findings will improve the reliability of gene expression research in *Reynoutria*, providing a foundation for future studies on the molecular regulation of secondary metabolism and stress responses in this ecologically and medicinally valuable plant group.

## 4. Materials and Methods

### 4.1. Plant Material

Three plant organs (R—rhizome, L—leaf, and FL—flower) from four representative specimens of three species—one *R. japonica* (RJ), two *R.* × *bohemica* (RB), and one *R. sachalinensis* (RS)—were collected in the urban environment of Wroclaw, Poland, from previously described locations: 5RJ, 13RB, 18RB, and 21RS [26], with all samples collected during the peak flowering season for each specimen. Voucher specimens of all taxa have been duly deposited at the Herbarium of the Botanical Garden of Medicinal Plants, Wroclaw Medical University (Poland). The plants used as experimental material in this study were collected by the staff of the botanical garden, following the decision of the Ministry of Environment, number: DOPozgiz-4210-26-6024-05/KL. The plants were formally identified by Klemens Jakubowski, MSc, curator with the Botanic Garden of Medicinal Plants. All samples were immediately frozen in liquid nitrogen, homogenized, and subsequently stored at −80 °C for future experiments.

### 4.2. RNA Extraction and cDNA Synthesis

Total RNA was extracted using the Plant/Fungi Total RNA Purification Kit (Norgen Biotek Corp., Thorold, Canada) from 50 mg of each sample in six replicates. RNA purification with DNase treatment (RNase-Free DNase I, Norgen Biotek Corp., Thorold, Canada) was performed using an on-column workflow with minor modifications to the manufacturer’s protocol. Since *Reynoutria* tissues are rich in polysaccharides, it was necessary to double the volume of Lysis Buffer to avoid column clogging, and the maximum recommended concentration of β-mercaptoethanol (2%) was used. The extracted RNA was assessed for quality and purity by the NanoDrop 2000 spectrophotometer (Thermo Fisher Scientific™, Waltham, MA, USA). RNA samples with an A260/A280 ratio of 1.9–2.1 and an A260/A230 ratio > 2.0 were used for subsequent cDNA synthesis. Between each successive step, the RNA was stored at –80 °C to maintain its quality and integrity.

cDNA was synthesized from 5 µg of total RNA in a total volume of 40 µL using the iScript™ Advanced cDNA Synthesis Kit for qRT-PCR (Bio-Rad, Hercules, CA, USA). The initial reaction comprised: 5 µg of total RNA, 8 µL of 5× iScript™ Advanced Reaction Mix (Bio-Rad, Hercules, CA, USA) (which includes dNTPs, oligo (dT), and random primers), 2 µL of iScript™ Advanced Reverse Transcriptase (Bio-Rad, Hercules, CA, USA), and nuclease-free water. This mixture was incubated at 46 °C for 20 min, followed by a heating step at 95 °C for 1 min. Subsequently, the cDNA was diluted to an ultimate volume of 80 µL with nuclease-free water (to 2.5 µg of total RNA input per sample).

### 4.3. Selection of Candidate Reference Genes and Primer Design

Twelve candidate reference genes—*ACT*, *TUA*, *TUB*, *GAPDH*, *EF-1γ*, *UBQ*, *UBC*, *60SrRNA*, *eIF6A*, *SKD1*, *YLS8,* and *NDUFA13* — were selected based on previous studies on *Reynoutria japonica* [31]. All primers proposed by Wang et al. were employed. Since those studies focused exclusively on *Polygonum cuspidatum* (*R. japonica*), additional transcriptomic data for *R. japonica* and *R. sachalinensis* were retrieved from the NCBI Sequence Read Archive (SRA) database (https://www.ncbi.nlm.nih.gov/sra/; accessed on 10 January 2025), as no transcriptomic data are available for the hybrids. Accession: PRJNA623335 data for *R. japonica* [54] and accession: PRJDB3954 for *R. sachalinensis* [55] were used. These data were assembled using Galaxy software version 24.2 (https://usegalaxy.eu/, accessed on 15 February 2025) [56]. The primers obtained from Wang et al. [31] were validated using the newly assembled transcriptome data, and new primer pairs were designed for both species using Primer3Plus version: 3.3.0 (https://www.primer3plus.com/; accessed on 8 March 2025) [57], assuming that the hybrid would respond to primers designed for the parent organisms.

Primer design parameters were set to a melting temperature of 60 °C, length of 20–22 bp, and ~50% GC content (Table 1). Prior to qRT-PCR assays, each primer pair’s amplification efficiency (E) and specificity were assessed using a ten-fold cDNA dilution series to generate standard curves. Efficiency was calculated as E [%] = (10^−1/slope^ − 1) × 100% [58].

The performance of our primers was then compared with that initially proposed by Wang et al. [31]. For each gene, the primer pair demonstrating the highest efficiency, specificity, and consistent amplification across all three *Reynoutria* species was selected.

### 4.4. Quantitative Real-Time PCR

Gene-specific primers (Table 1) were employed to quantify transcript levels from reverse-transcribed cDNA templates. Reactions were carried out in triplicate with a TOptical Gradient 96 system (Biometra Analytic, Jena, Germany) using SsoAdvanced™ Universal SYBR Green Supermix^®^ (Bio-Rad, Hercules, CA, USA). The reaction mixture consisted of 5 μL of SsoAdvanced Universal SYBR^®^ Green Supermix (2x) (Bio-Rad, Hercules, CA, USA), 2 μL of nuclease-free H_2_O, 1 μL of previously diluted cDNA, and 2 μL of a mix of forward and reverse amplification primers (250 nmol/μL each), resulting in a final volume of 10 μL.

Thermal cycling comprised an initial denaturation at 95 °C for 30 s, followed by 40 cycles of 95 °C for 3 s and 60 °C for 30 s. A subsequent melt-curve analysis was performed from 60 °C to 95 °C at a rate of 1 °C per 6 s to verify amplicon specificity. Primer efficiencies were determined experimentally from a ten-fold cDNA dilution series and incorporated into relative quantification calculations via the Pfaffl method [59]. All stages, from experimental design and RNA isolation through data analysis, complied with the MIQE guidelines for qRT-PCR [60]. Relative expression values are provided in Appendix A.

### 4.5. Statistical Data Analysis

Several commonly used algorithms, including ΔCt, BestKeeper, GeNorm, NormFinder, and a web-based tool RefFinder 2014 Release (original version) (https://www.ciidirsinaloa.com.mx/RefFinder-master/; accessed on 20 May 2025), were applied to assess the stability of candidate reference genes [4,52,61] in various tissues of closely related *Reynoutria* species. GeNorm and NormFinder use the 2^−ΔCT^ method for calculations, while BestKeeper analyzes the Ct values. GeNorm calculates the stability of a reference gene (M) based on the average pairwise variations (V) among all other reference genes. It also determines the optimal number of reference genes needed by computing the V_n_/V_n+1_ pairwise variation. The NormFinder algorithm evaluates both intra- and intergroup expression variations, assigning the highest stability to genes with the lowest stability values (SV). The BestKeeper program evaluates the expression stability of genes by assessing coefficients of variation (CV) and standard deviations (SD), with the lowest CV and SD used as criteria for identifying the most stable reference genes. Lastly, the results from the above algorithms are combined by the web-based analysis tool RefFinder. RefFinder is an online tool that brings together the results of major computational programs, including geNorm (M values), NormFinder (stability values), BestKeeper (CV and SD), and ΔCt values. It calculates the overall expression stability ranking by taking the geometric mean of all rankings, which helps choose the most suitable reference genes.

### 4.6. Validation of Selected Reference Genes

Using the selected reference genes, we further analyzed the expression patterns of *chalcone synthase* (*CHS*) and *phenylalanine ammonia-lyase* (*PAL*) across various tissues in three species to validate the stability of these reference genes. The samples and procedures remained the same as those used in the qPCR analysis of the reference genes. We assessed the relative expression level using the calculated primer efficiency [59].

## 5. Conclusions

In this study, the stability of twelve candidate reference genes was assessed using four commonly applied algorithms: ΔCt, geNorm, NormFinder, and BestKeeper, with results comprehensively integrated by RefFinder. The combined analysis identified *SKD1*, *60SrRNA*, and *ACT* as the most stable and reliable reference genes for interspecies normalization across *Reynoutria* species. Species-specific analysis showed that *SKD1* and *EF-1γ* were optimal for *R. japonica*; *UBC*, *eIF6A*, *60SrRNA*, and *ACT* for *R. × bohemica*; and *SKD1*, *60SrRNA*, and *ACT* for *R. sachalinensis*. Organ-specific analysis further revealed that *60SrRNA* and *EF-1γ* were the most stable in leaves; *SKD1*, *60SrRNA*, *NDUFA13*, and *ACT* in rhizomes; and *SKD1* and *TUB* in flowers. Validation using *CHS* and *PAL* expression patterns confirmed that normalization with *SKD1*, *60SrRNA*, and *ACT* yields consistent and biologically meaningful results. This study provides a valuable foundation for accurate gene expression analysis in *Reynoutria* and supports future transcriptomic research in these closely related species.

## Figures and Tables

**Figure 1 ijms-26-08265-f001:**
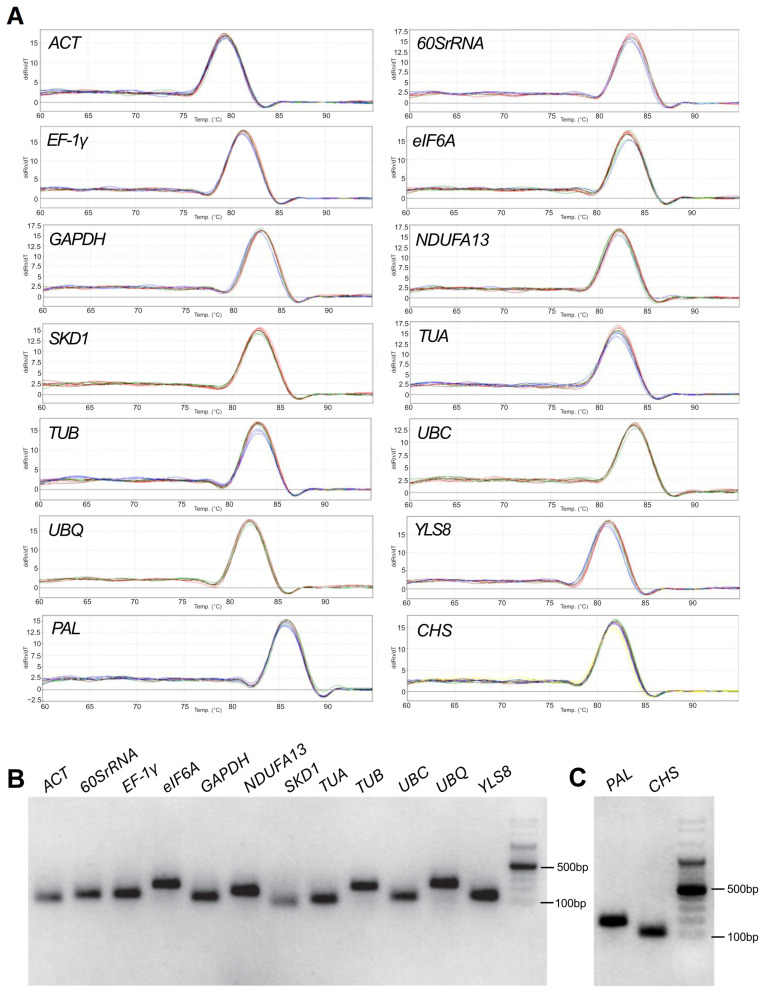
Primer specificity of candidate reference genes verified by melting curve analysis (**A**) and 1.5% agarose gel electrophoresis (**B**,**C**). Amplicon sizes were estimated against the mass marker Perfect™ 100 bp DNA Ladder (EURx, Gdansk, Poland).

**Figure 2 ijms-26-08265-f002:**
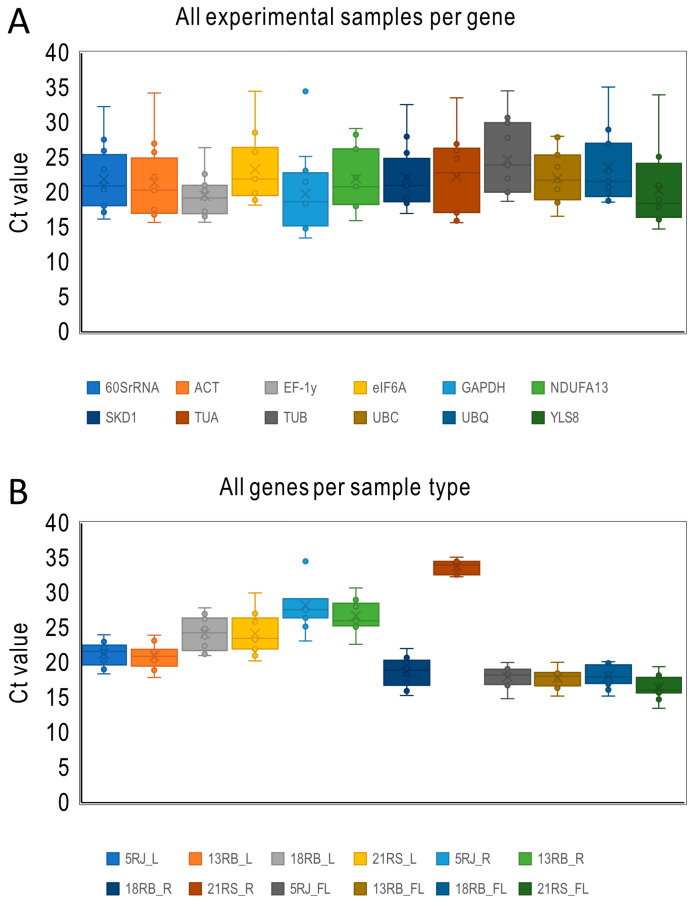
Expression levels of 12 candidate reference genes across three tissues of three *Reynoutria* species, presented as cycle threshold (Ct) values. Distribution of mean Ct values of 12 experimental samples per gene (**A**) and distribution of mean Ct values of 12 genes per experimental sample (**B**). For each boxplot, the average (diagonal cross), median (horizontal line), interquartile range (boxes: 25th to 75th percentile), and range (whiskers: minimum to maximum) are displayed, illustrating variation in transcript abundance across the genes (**A**) or samples (**B**).

**Figure 3 ijms-26-08265-f003:**
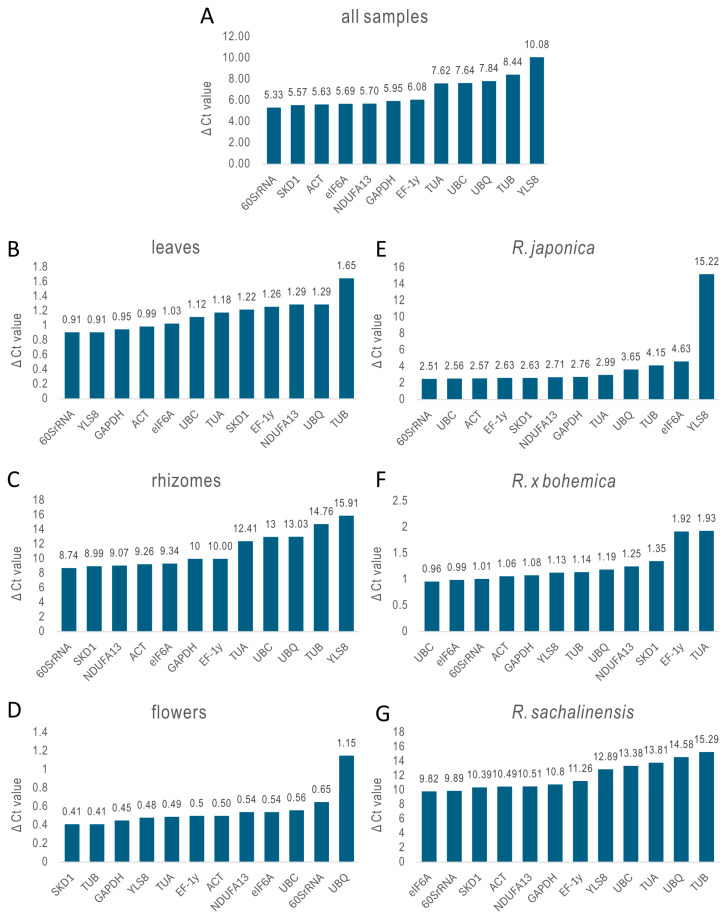
Stability evaluation of 12 reference genes analyzed using ∆Ct. All samples (**A**), leaves of three species (**B**), rhizomes of three species (**C**), flowers of three species (**D**), three organs of *R. japonica* (**E**), three organs of *R. × bohemica* (**F**), and three organs of *R. sachalinensis* (**G**).

**Figure 4 ijms-26-08265-f004:**
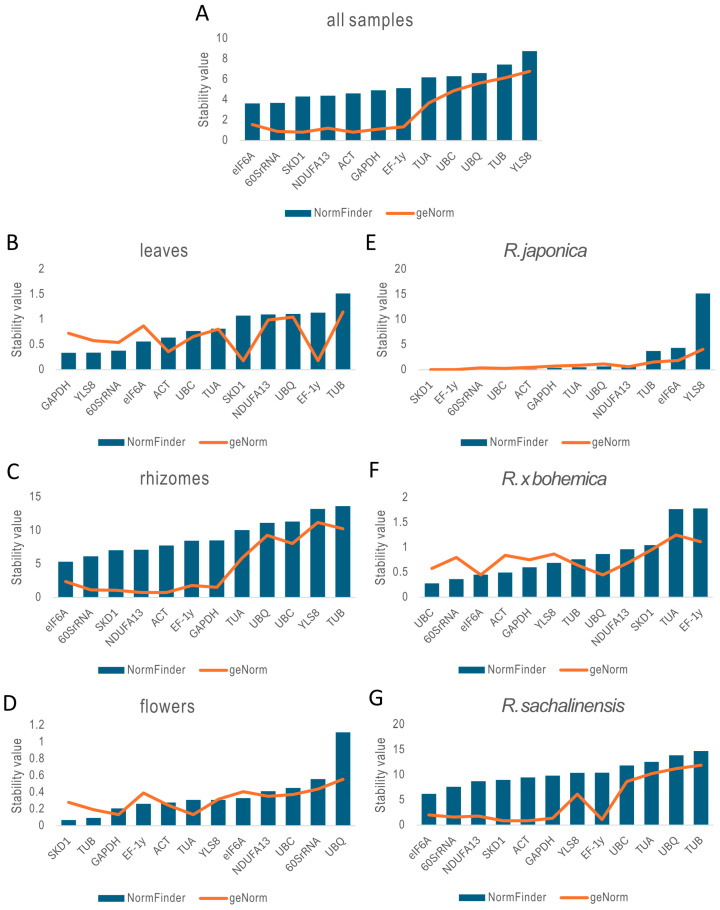
The ranking order of reference genes as expression stability values (M) of candidate reference genes calculated by geNorm (orange line) and NormFinder (blue bars). All samples (**A**), leaves of three species (**B**), rhizomes of three species (**C**), flowers of three species (**D**), three organs of *R. japonica* (**E**), three organs of *R.* × *bohemica* (**F**), and three organs of *R. sachalinensis* (**G**).

**Figure 5 ijms-26-08265-f005:**
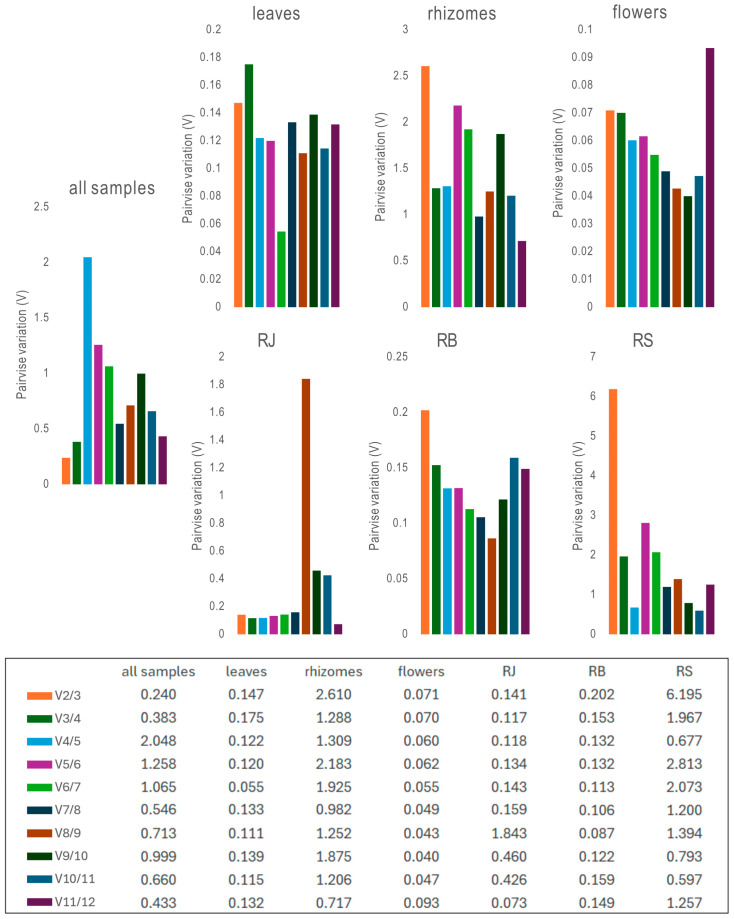
Pairwise variation (V) of 12 candidate reference genes calculated by geNorm. V_n_/V_n+1_ values were used to determine the optimal number of reference genes.

**Figure 6 ijms-26-08265-f006:**
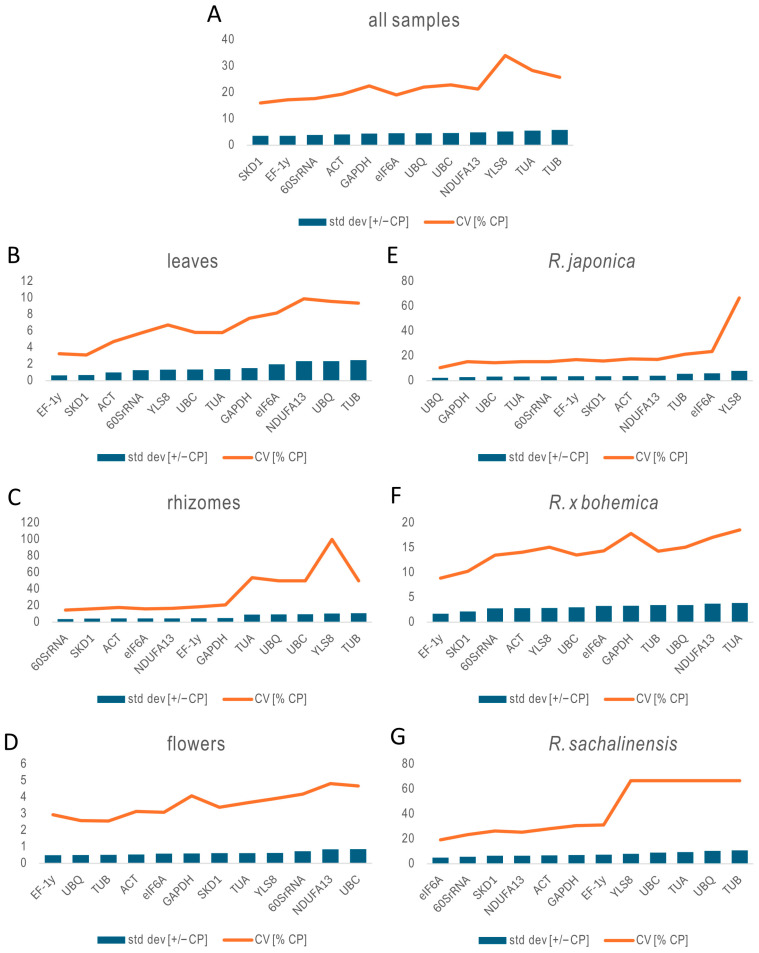
Stability evaluation of reference genes calculated by BestKeeper. All samples (**A**), leaves of three species (**B**), rhizomes of three species (**C**), flowers of three species (**D**), three organs of *R. japonica* (**E**), three organs of *R. × bohemica* (**F**), and three organs of *R. sachalinensis* (**G**).

**Figure 7 ijms-26-08265-f007:**
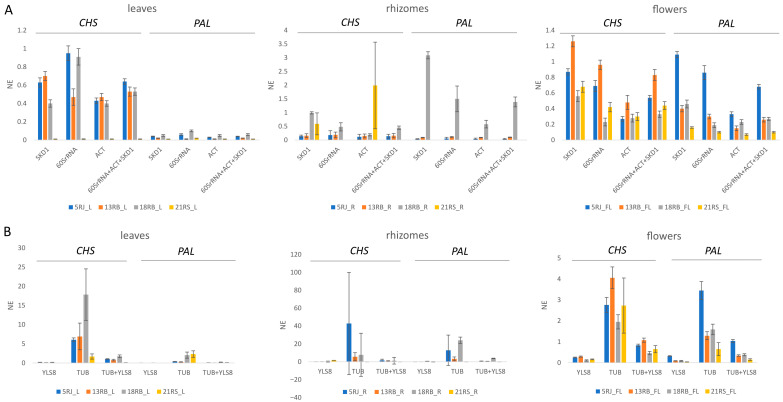
Relative expression levels of target genes CHS and PAL in leaves, rhizomes, and flowers of three *Reynoutria* species. The most stable reference genes (*SKD1*, *60SrRNA,* and *ACT*), alone and in combination (**A**), and the most unstable genes (*YLS8* and *TUB*) (**B**) were selected as the normalization factors. Error bars show the standard error calculated from three biological replicates.

**Table 1 ijms-26-08265-t001:** qPCR primers used in this study.

Primer Name	Primer Sequence 5′ —> 3′	Target Based on Alignment Analysis	Product Length [bp]	Efficiency [%]	R^2^	Reference	Accession Number
*Actin 7 (ACT)*							
actF1	GCCGTCTATGATTGGAATGG	*Reynoutria* spp.	99	104	0.998	[31]	MK288156
actR1	TACCGTACAAGTCCTTCCTAA	*Reynoutria* spp.				[31]	

*Tubulin-alpha 6 (TUA)*							
tuaF2	AGGGCCGTTTGCATGATTTC	*Reynoutria* spp.	101	101	0.996	This study	MK288157
tuaR2	TGAACAAAGGCACGCTTAGC	*Reynoutria* spp.				This study	

*Tubulin-beta 2 (TUB)*							
tubF1	ATCCGACACTGTTGTTGAGC	*R. japonica/Reynoutria* spp.	211	102	0.999	[31]	MK288158
tubR2	TTGGCCAGGGAAACGTAAAC	*Reynoutria* spp.				This study	

*Glyceraldehyde-3-phosphate dehydrogenase (GAPDH)*							
gapdhF2	TGACCACAGTTCACGCAATG	*Reynoutria* spp.	108	101	0.999	This study	MK288159
gapdhR2	TGCTGCTGGGAATGATGTTG	*Reynoutria* spp.				This study	

*Elongation factor 1-gamma (EF-1γ)*							
ef1γF2	AGCCGCATCATGACCAAAAC	*Reynoutria* spp.	131	101	0.998	This study	MK288160
ef1γR2	ATTGCTGGAACGGATGTTGC	*Reynoutria* spp.				This study	

*Ubiquitin domain-containing protein (UBQ)*							
ubqF2	ATTGGAGCAGATGCAGCAAC	*Reynoutria* spp.	240	100	0.999	This study	MK288161
ubqR2	ATTTCACGCATGAGCTCTGG	*Reynoutria* spp.				This study	

*Ubiquitin-conjugating enzyme (UBC)*							
ubcF1	ATTTGATGGCGTGGAGTTGC	*R. japonica*	119	101	0.998	[31]	MK288162
ubcR2	TTTACACTTTGGGGGCTTGC	*Reynoutria* spp.				This study	

*60S ribosomal RNA (60SrRNA)*							
60SF	ACTGTGATTTCGCAGACGCA	*R. japonica*	124	101	0.997	[31]	MK288163
60SR	CCTGGTGCTTGGTGAGACGG	*R. japonica*				[31]	

*Eukaryotic translation initiation factor 6A (eIF6A)*							
elf6aF2	TGGTTGCAATTGGTGGATCC	*Reynoutria* spp.	220	100	0.99	This study	MK288164
elf6aR2	TCAATGCGCTGGACAACAAC	*Reynoutria* spp.				This study	

*Suppressor of K+ transport growth defect 1 (SKD1)*							
skdF2	AGAAGCCGAATGTGAAGTGG	*Reynoutria* spp.	75	102	0.994	This study	MK288165
skdR2	ATATGACCGCTTCCTGCAAC	*Reynoutria* spp.				This study	

*Thioredoxin-like protein YLS8 (YLS8)*							
ylsF2	TGCCCGACTTCAACACAATG	*Reynoutria* spp.	138	102	0.999	This study	MK288166
ylsR2	ACTCCTGCTTGTCCTTTAGAGC	*Reynoutria* spp.				This study	

*NADH dehydrogenase [ubiquinone] 1 alpha subcomplex subunit 13-A (NDUFA13)*							
ndufaF1	ATGTACCAGGTCGGCGTAGG	*R. japonica/Reynoutria* spp.	161	104	0.999	[31]	MK288167
ndufaR1	TCCTTCATAATTCTGGCTTCC	*R. japonica/Reynoutria* spp.				[31]	

*Phenylalanine ammonia lyase (PAL)*							
palF	TATTGTCTGTCGGCGTCAAC	*Reynoutria* spp.	187	104	0.997	This study	MK288155
palR	TTCTCCTTGTCGCCGTTTTC	*Reynoutria* spp.				This study	

*Chalcone synthase (CHS)*							
chsF	AAACATGTCGAGTGCGTGTG	*Reynoutria* spp.	108	105	0.999	This study	MT415958
chsR	AACAAAACGCCCCACTCAAG	*Reynoutria* spp.				This study	

**Table 2 ijms-26-08265-t002:** Comprehensive stability rankings of candidate reference genes as determined by RefFinder.

Method	Ranking Order (from the Most Stable to the Least Stable Gene)											
	1	2	3	4	5	6	7	8	9	10	11	12
all samples												
Delta CT	*60SrRNA*	*SKD1*	*ACT*	*eIF6A*	*NDUFA13*	*GAPDH*	*EF-1γ*	*TUA*	*UBC*	*UBQ*	*TUB*	*YLS8*
BestKeeper	*SKD1*	*EF-1γ*	*60SrRNA*	*ACT*	*GAPDH*	*eIF6A*	*UBQ*	*UBC*	*NDUFA13*	*YLS8*	*TUA*	*TUB*
NormFinder	*eIF6A*	*60SrRNA*	*SKD1*	*NDUFA13*	*ACT*	*GAPDH*	*EF-1γ*	*TUA*	*UBC*	*UBQ*	*TUB*	*YLS8*
geNorm	*ACT|SKD1*		*60SrRNA*	*GAPDH*	*NDUFA13*	*EF-1γ*	*eIF6A*	*TUA*	*UBC*	*UBQ*	*TUB*	*YLS8*
Comprehensive ranking	*SKD1*	*60SrRNA*	*ACT*	*eIF6A*	*EF-1γ*	*GAPDH*	*NDUFA13*	*TUA*	*UBC*	*UBQ*	*TUB*	*YLS8*
leaves												
Delta CT	*60SrRNA*	*YLS8*	*GAPDH*	*ACT*	*eIF6A*	*UBC*	*TUA*	*SKD1*	*EF-1γ*	*NDUFA13*	*UBQ*	*TUB*
BestKeeper	*EF-1γ*	*SKD1*	*ACT*	*60SrRNA*	*YLS8*	*UBC*	*TUA*	*GAPDH*	*eIF6A*	*UBQ*	*NDUFA13*	*TUB*
NormFinder	*GAPDH*	*YLS8*	*60SrRNA*	*eIF6A*	*ACT*	*UBC*	*TUA*	*SKD1*	*NDUFA13*	*UBQ*	*EF-1γ*	*TUB*
geNorm	*EF-1γ|SKD1*		*ACT*	*60SrRNA*	*YLS8*	*UBC*	*GAPDH*	*TUA*	*eIF6A*	*NDUFA13*	*UBQ*	*TUB*
Comprehensive ranking	*60SrRNA*	*EF-1γ*	*YLS8*	*SKD1*	*GAPDH*	*ACT*	*UBC*	*eIF6A*	*TUA*	*NDUFA13*	*UBQ*	*TUB*
rhizomes												
Delta CT	*60SrRNA*	*SKD1*	*NDUFA13*	*ACT*	*eIF6A*	*GAPDH*	*EF-1γ*	*TUA*	*UBC*	*UBQ*	*TUB*	*YLS8*
BestKeeper	*60SrRNA*	*SKD1*	*ACT*	*eIF6A*	*NDUFA13*	*EF-1γ*	*GAPDH*	*TUA*	*UBQ*	*UBC*	*YLS8*	*TUB*
NormFinder	*eIF6A*	*60SrRNA*	*SKD1*	*NDUFA13*	*ACT*	*EF-1γ*	*GAPDH*	*TUA*	*UBQ*	*UBC*	*YLS8*	*TUB*
geNorm	*ACT|NDUFA13*		*SKD1*	*60SrRNA*	*GAPDH*	*EF-1γ*	*eIF6A*	*TUA*	*UBC*	*UBQ*	*TUB*	*YLS8*
Comprehensive ranking	*60SrRNA*	*SKD1*	*NDUFA13*	*ACT*	*eIF6A*	*GAPDH*	*EF-1γ*	*TUA*	*UBC*	*UBQ*	*TUB*	*YLS8*
flowers												
Delta CT	*SKD1*	*TUB*	*GAPDH*	*YLS8*	*TUA*	*EF-1γ*	*ACT*	*NDUFA13*	*eIF6A*	*UBC*	*60SrRNA*	*UBQ*
BestKeeper	*EF-1γ*	*UBQ*	*TUB*	*ACT*	*eIF6A*	*GAPDH*	*SKD1*	*TUA*	*YLS8*	*60SrRNA*	*NDUFA13*	*UBC*
NormFinder	*SKD1*	*TUB*	*GAPDH*	*EF-1γ*	*ACT*	*TUA*	*YLS8*	*eIF6A*	*NDUFA13*	*UBC*	*60SrRNA*	*UBQ*
geNorm	*GAPDH|TUA*		*TUB*	*ACT*	*SKD1*	*YLS8*	*NDUFA13*	*UBC*	*EF-1γ*	*eIF6A*	*60SrRNA*	*UBQ*
Comprehensive ranking	*SKD1*	*TUB*	*GAPDH*	*EF-1γ*	*TUA*	*ACT*	*YLS8*	*UBQ*	*eIF6A*	*NDUFA13*	*UBC*	*60SrRNA*
*R. japonica*												
Delta CT	*60SrRNA*	*UBC*	*ACT*	*EF-1γ*	*SKD1*	*NDUFA13*	*GAPDH*	*TUA*	*UBQ*	*TUB*	*eIF6A*	*YLS8*
BestKeeper	*UBQ*	*GAPDH*	*UBC*	*TUA*	*60SrRNA*	*EF-1γ*	*SKD1*	*ACT*	*NDUFA13*	*TUB*	*eIF6A*	*YLS8*
NormFinder	*SKD1*	*EF-1γ*	*60SrRNA*	*UBC*	*ACT*	*GAPDH*	*TUA*	*UBQ*	*NDUFA13*	*TUB*	*eIF6A*	*YLS8*
geNorm	*EF-1γ|SKD1*		*UBC*	*60SrRNA*	*ACT*	*NDUFA13*	*GAPDH*	*TUA*	*UBQ*	*TUB*	*eIF6A*	*YLS8*
Comprehensive ranking	*SKD1*	*EF-1γ*	*60SrRNA*	*UBC*	*GAPDH*	*ACT*	*UBQ*	*TUA*	*NDUFA13*	*TUB*	*eIF6A*	*YLS8*
*R. × bohemica*												
Delta CT	*UBC*	*eIF6A*	*60SrRNA*	*ACT*	*GAPDH*	*YLS8*	*TUB*	*UBQ*	*NDUFA13*	*SKD1*	*EF-1γ*	*TUA*
BestKeeper	*EF-1γ*	*SKD1*	*60SrRNA*	*ACT*	*YLS8*	*UBC*	*eIF6A*	*GAPDH*	*TUB*	*UBQ*	*NDUFA13*	*TUA*
NormFinder	*UBC*	*60SrRNA*	*eIF6A*	*ACT*	*GAPDH*	*YLS8*	*TUB*	*UBQ*	*NDUFA13*	*SKD1*	*TUA*	*EF-1γ*
geNorm	*eIF6A|UBQ*		*UBC*	*TUB*	*NDUFA13*	*GAPDH*	*60SrRNA*	*ACT*	*YLS8*	*SKD1*	*EF-1γ*	*TUA*
Comprehensive ranking	*UBC*	*eIF6A*	*60SrRNA*	*ACT*	*UBQ*	*GAPDH*	*EF-1γ*	*YLS8*	*TUB*	*SKD1*	*NDUFA13*	*TUA*
*R. sachalinensis*												
Delta CT	*eIF6A*	*60SrRNA*	*SKD1*	*ACT*	*NDUFA13*	*GAPDH*	*EF-1γ*	*YLS8*	*UBC*	*TUA*	*UBQ*	*TUB*
BestKeeper	*eIF6A*	*60SrRNA*	*SKD1*	*NDUFA13*	*ACT*	*GAPDH*	*EF-1γ*	*YLS8*	*UBC*	*TUA*	*UBQ*	*TUB*
NormFinder	*eIF6A*	*60SrRNA*	*NDUFA13*	*SKD1*	*ACT*	*GAPDH*	*YLS8*	*EF-1γ*	*UBC*	*TUA*	*UBQ*	*TUB*
geNorm	*ACT|SKD1*		*EF-1γ*	*GAPDH*	*60SrRNA*	*NDUFA13*	*eIF6A*	*YLS8*	*UBC*	*TUA*	*UBQ*	*TUB*
Comprehensive ranking	*eIF6A*	*SKD1*	*60SrRNA*	*ACT*	*NDUFA13*	*GAPDH*	*EF-1γ*	*YLS8*	*UBC*	*TUA*	*UBQ*	*TUB*

## Data Availability

Data generated and analyzed during this study are included in this published article and its Appendix A.

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
