# Peer review of "Stabilizing the Baseline: Reference Gene Evaluation in Three Invasive Reynoutria Species"

_ijms, 2025, doi:10.3390/ijms26178265_

Round 1
Reviewer 1 Report
Comments and Suggestions for Authors
The research is a systematic, comparative validation of HKGs across
closely related knotweed species (Reynoutria), providing a robust foundation for future transcriptomic and functional studies of their specialized metabolism and other biological processes. The objects of the study are R. japonica Houtt., R. sachalinensis (F. Schmidt) Nakai, and their hybrid R. × bohemica Chrtek & Chrtková. These three species are among the most aggressive invasive plants worldwide. Rhizomes, leaves and flowers from the three species are used for the analyses. The design of the experiment is appropriate, standard methods are used and comprehensively described. The results are scientifically sound and the discussion is convincing.
There are few minor comments
The charts in fig 7 are too small to read. There should be ax two charts on a line so that they are bigger but better make 6 rows of chars and just one chart on a row.
Please provide identification key based on morphological features. That will be very useful to distinguish morphologically the three taxa. It will bring additional high citation of the paper.
Author Response
Dear Reviewer,
We sincerely thank you for taking the time to review our manuscript and for providing constructive and insightful comments. Your feedback has been invaluable in improving the quality and clarity of our work. We have carefully considered each suggestion, revised the manuscript accordingly, and highlighted all changes using the track changes function. Detailed point-by-point responses to all issues raised are provided below, with our answers outlined in bold corresponding directly to the reviewers’ remarks.
Comments 1: The charts in fig 7 are too small to read. There should be ax two charts on a line so that they are bigger but better make 6 rows of chars and just one chart on a row.
Response 1: We sincerely thank the reviewer for this valuable suggestion. We agree that increasing the chart size improves readability. We tested both suggested options and noticed that the version with one chart in a row does not enlarge the size of diagrams when scaled down to fit the figure into the A4 format. The version with two charts in a row fits nicely into the A4 format, but does not allow for organizing diagrams into two panels A and B. This version is, however, equal to the original Figure 7 if placed in the landscape orientation. Accordingly, we have reformatted Figure 7 to ensure a larger and more precise visualization of the data. Hopefully, the reviewer will find this solution satisfying. If not, we are open to further suggestions.
Comments 2: Please provide identification key based on morphological features. That will be very useful to distinguish morphologically the three taxa. It will bring additional high citation of the paper.
Response 2: We appreciate the insightful suggestion regarding the inclusion of a morphological identification key. Indeed, distinguishing between Reynoutria species and hybrids is very important; hence, our previous publication was dedicated entirely to this issue (DOI: 10.1038/s41598-025-90494-2). The publication provided a comprehensive key for distinguishing Reynoutria japonica, R. sachalinensis, and R. × bohemica based on morphological features, metabolic profiles, and molecular markers. Since this work is directly cited and discussed in the present manuscript – citation number [26], we believe it adequately serves this purpose while avoiding unnecessary repetition. However, a short mention about this has been added to the Introduction (lines 78-82).
Reviewer 2 Report
Comments and Suggestions for Authors
- What is the significance of this study? It is necessary to explain in detail that there are many ways to detect invasion of foreign species and generate gene flow. What are the advantages of choosing this gene?
- In the stability analysis part of gene expression, there are five parts of content. There are too many tables and Fig need to be streamlined and the main results are written.
- 2 genes were selected as marker genes, Fig 7 is unclear and needed improvement.
Author Response
Dear Reviewer,
We are very grateful for your careful evaluation of our manuscript and your thoughtful suggestions for improvement. In revising the manuscript, we have addressed all comments in detail and implemented the recommended changes to enhance the clarity, readability, and overall quality of the work. All revisions have been marked using the track changes option for transparency. A detailed point-by-point response to the reviewer’s comments is provided below, with our answers highlighted in bold and references to the specific modifications made in the revised manuscript.
Comments 1: What is the significance of this study? It is necessary to explain in detail that there are many ways to detect invasion of foreign species and generate gene flow. What are the advantages of choosing this gene?
Response 1: We thank the reviewer for highlighting this critical point. In our previous work (DOI: 10.1038/s41598-025-90494-2), we addressed the invasiveness, population structure, and species identification of Reynoutria spp. and proposed molecular markers to track these processes. Our results highlighted that morphologically similar plants may differ considerably at the genetic and phytochemical levels, underscoring the importance of molecular characterization to avoid misidentification and the associated risks of unexpected pharmacological properties. The present study has a different but complementary focus. Here, the analysis is centered on the stability of reference genes that can be reliably used in transcriptomic analyses. This is a critical step for future research on the regulation of biosynthetic pathways of bioactive compounds, since Reynoutria japonica is not only an invasive species but also a pharmacopoeial plant widely used in herbal medicine.
This study presents a first approach to find reliable reference genes for comparative gene expression studies of Reynoutria spp. plants, both wildly growing and in experimental studies.
Since our previous work is directly cited and discussed in the present manuscript – citation number [26], we believe it adequately serves this purpose while avoiding unnecessary repetition. However, to clarify this, we have added a short mention in the Introduction of the revised manuscript (lines 83-88).
Comments 2: In the stability analysis part of gene expression, there are five parts of content. There are too many tables and Fig need to be streamlined and the main results are written
Response 2: We sincerely appreciate the reviewer’s insightful observation. In investigations of this nature, the scope of analyses frequently requires a strategic presentation of results. Following similar publications (DOI: https://doi.org/10.1038/s41598-024-51562-1, https://doi.org/10.1186/s12870-024-04924-w, https://doi.org/10.3390/ijms25053029), we have structured the Expression Stability Assessment to present the results of partial analyses, including those obtained from the ΔCt Method, geNorm, NormFinder, and BestKeeper. To improve readability and streamline the manuscript, some findings have already been combined into consolidated figures (e.g., Figure 4 and Figure 6) or tables (Table 2). At the same time, additional data are exclusively supplied in the Supplementary Materials. Omitting specific partial data could give the impression of missing information; therefore, we considered it essential to present these results in the proposed format.
Main results are presented in the Comprehensive Stability Ranking paragraph and highlighted in bold in Table 2. The significance of the principal findings has been underscored in the concluding paragraph of the Discussion. We welcome any additional suggestions that could enhance the transparency in presenting our primary results.
Comments 3: 2 genes were selected as marker genes, Fig 7 is unclear and needed improvement.
Response 3: We appreciate the reviewer's valuable comment. In similar studies, it is common practice to select 1-3 marker genes to illustrate the activity of the proposed housekeeping genes. Comparable investigations into reference gene selection and stability can be found in recent works (DOI: 10.3390/ijms242015087, https://doi.org/10.1002/fsn3.4765, https://www.nature.com/articles/s41598-025-92244-w). Hence, we have selected two marker genes, PAL and CHS, which are crucial for the phenylpropanoid and flavonoid biosynthesis pathways, respectively, and play key roles in plant defense and the production of secondary metabolites such as stilbenes and vanicosides – compounds of particular significance in Reynoutria species. We have further analyzed the expression of PAL and CHS across different tissues of Reynoutria when normalized using the stable (panel A of Fig. 7) and unstable (panel B of Fig. 7) housekeeping genes, thus demonstrating their usefulness as a reference for qRT-PCR studies.
We acknowledge that the original presentation of Figure 7 was difficult to read, and we have reformatted it in accordance with the reviewer’s suggestion. We have tested several options and decided that landscape orientation provides a larger and more precise visualization of the Figure 7 data, improving its readability. Hopefully, the reviewer will find this solution satisfying. If not, we are open to further suggestions.
Round 2
Reviewer 2 Report
Comments and Suggestions for Authors
No